# Transfer of Antibiotic Resistance Plasmid from Commensal *E. coli* towards Human Intestinal Microbiota in the M-SHIME: Effect of *E. coli* dosis, Human Individual and Antibiotic Use

**DOI:** 10.3390/life11030192

**Published:** 2021-02-28

**Authors:** Ellen Lambrecht, Els Van Coillie, Nico Boon, Marc Heyndrickx, Tom Van de Wiele

**Affiliations:** 1Center for Microbial Ecology and Technology (CMET), Ghent University, Coupure Links 653, 9000 Gent, Belgium; ellen.lambrecht@ilvo.vlaanderen.be (E.L.); nico.boon@ugent.be (N.B.); 2Flanders Research Institute for Agriculture (ILVO), Fisheries and Food, Brusselsesteenweg 370, 9090 Melle, Belgium; els.vancoillie@ilvo.vlaanderen.be (E.V.C.); marc.heyndrickx@ilvo.vlaanderen.be (M.H.); 3Department of Pathology, Bacteriology and Avian Diseases, Ghent University, Salisburylaan 133, 9820 Merelbeke, Belgium

**Keywords:** M-SHIME, antibiotic resistance, resistance transfer

## Abstract

Along with (in)direct contact with animals and a contaminated environment, humans are exposed to antibiotic resistant bacteria by consumption of food. The implications of ingesting antibiotic resistant commensal bacteria are unknown, as dose-response data on resistance transfer and spreading in our gut is lacking. In this study, transfer of a resistance plasmid (IncF), harbouring several antibiotic resistance genes, from a commensal *E. coli* strain towards human intestinal microbiota was assessed using a Mucosal Simulator of the Human Intestinal Ecosystem (M-SHIME). More specifically, the effect of the initial *E. coli* plasmiddonor concentration (10^5^ and 10^7^ CFU/meal), antibiotic treatment (cefotaxime) and human individual (n = 6) on plasmid transfer towards lumen coliforms and anaerobes was determined. Transfer of the resistance plasmid to luminal coliforms and anaerobes was observed shortly after the donor strain arrived in the colon and was independent of the ingested dose. Transfer occurred in all six simulated colons and despite their unique microbial community composition, no differences could be detected in antibiotic resistance transfer rates between the simulated human colons. After 72 h, resistant coliform transconjugants levels ranged from 7.6 × 10^4^ to 7.9 × 10^6^ CFU_cefotaxime resistant_/mL colon lumen. Presence of the resistance plasmid was confirmed and quantified by PCR and qPCR. Cefotaxime treatment led to a significant reduction (85%) in resistant coliforms, however no significant effect on the total number of cultivable coliforms and anaerobes was observed.

## 1. Introduction

Over- and misuse of antibiotics in clinical and veterinary medicine has contributed to the emergence and spread of antibiotic resistance genes [1]. These genes are often located on mobile genetic elements such as plasmids, transposons, integrons, integrative conjugative elements and genomic islands, and can be disseminated by horizontal gene transfer [2]. Antimicrobial resistance in pathogenic bacteria as well as commensals is an emerging public health threat, especially since transfer of antimicrobial resistance genes among pathogens or from commensals to pathogens can occur [3]. Antibiotic resistance genes are widespread and are circulating among bacteria in natural and anthropogenic environments, including the animal and human gastrointestinal tract [2,4,5], creating a continuous resistance gene flow [6]. Comparative genome analyses have illustrated that inter-environmental resistance transfer between animals and humans occurs at higher ratios than those from aquatic or terrestrial environment towards humans, depicting a contributing role of animals in disseminating antibiotic resistance genes to humans [4,5]. Whether food consumption is the main factor for this resistance dissemination to humans is not clear, as many studies use traditional typing methods which often have insufficient resolution to reliably assess the strain relatedness and the dissemination of resistant bacteria from the food chain to humans. Nonetheless, a recent study reported a median antibiotic resistant bacteria prevalence level of >50% on retail food samples in Europe [7]. De Been and colleagues provided strong evidence based on whole genome data for the clonal transfer of an ESBL producing *E. coli* between pigs and pig farmers, which may have occurred by direct contact, aerosols or (cross-contaminated) food [8]. Together with the global increase of fecal colonization of resistant Enterobacteriaceae in humans and animals [9], this implies that resistant bacteria can colonize our gut and may also transfer their antibiotic resistance genes towards human gut microbiota. The human gut harbors a diverse and dense bacterial community which has profound effects in human health and physiology [10]. Despite its low phylum level diversity, with Firmicutes, Bacteroidetes, Actinobacteria, and Proteobacteria being the dominant ones, it is characterized by a high species richness [11] and is often considered as a hotspot for gene transfer [5]. Transfer of antibiotic resistance genes mainly occurs within a genus, but also between phylogenetically distant species, since some genes were even found to be transferred across five phyla or more [5]. Mobile ARGs are mainly present in Proteobacteria, Firmicutes, Bacteroidetes and Actinobacteria and are significantly enriched in Proteobacteria, more specifically in *E. coli, Pseudomonas aeruginosa, Klebsiella pneumoniae, Klebsiella oxytoca and Enterobacter cloacae* [4]. In addition, genome sequencing revealed that *E. coli* and *K. pneumoniae* shared the largest number of mobile resistance genes [5].

In vivo resistance transfer events between Enterobacteriaceae have been observed in the gut of antibiotic treated, hospitalized infants [12,13,14] and in those of elderly persons [15]. Antibiotic treatment generates a selective pressure which can drive resistance development and transfer [16]. However, the transfer of a conjugative plasmid harboring antibiotic resistance genes from an ingested bacterium was also observed in a (simulated) human gut in the absence of antibiotics [17,18].

The daily human exposure levels to commensal antibiotic resistant bacteria are not known yet. Human exposure assessment models estimated exposure levels in the range of 1 × 10^−2^ and 1.35 × 10^6^ CFU resistant *E. coli* per 100 g lettuce [19] and a 1.5% chance to be exposed to more than 1000 CFU cephalosporin resistant *E. coli* after consumption of a chicken meal [20]. Moreover, there is a paucity of information about the amount of these bacteria needed to initiate resistance transfer and the frequency and speed at which transfer to gut microbiota takes place. 

This study aimed to evaluate the effect of (i) the ingested dose of resistant *E. coli*, (ii) the human individual and (iii) antibiotic treatment, on resistance plasmid transfer towards gut microbiota by using an in vitro model mimicking the human intestinal system. More specifically, transfer of a IncFII-resistance plasmid, harbouring several antibiotic resistance genes, from a commensal *E. coli* originating from a broiler chicken towards colon coliforms and anaerobes was determined in a Mucosal Simulator of the Human Intestinal Ecosystem (M-SHIME) setup from stool of 6 different human individuals. 

## 2. Materials and Methods

### 2.1. E. coli MB6212 Plasmid Donor Strain

The commensal *Escherichia coli* strain, MB6212, isolated from a broiler and known to transfer in vitro [17] its resistance plasmid (p5876, accession number MK070495, Appendix A) was selected for this study. The strain had been made lactose-negative to track plasmid transfer to indigenous coliforms in the M-SHIME [17]. *E. coli* MB6212 (the plasmid donor) grows as white colonies on MacConkey plates (Oxoid), whereas transconjugants (i.e., indigenous coliforms which accepted the plasmid) form red colonies. Moreover, the lactose negative MB6212 appears as pink colonies on RAPID’E. coli Agar (Biorad), whereas the wild type strain and other innate *E. coli* forms purple colonies, favoring separate enumerations. The MB6212 strain was grown overnight in Tryptone Soy Broth (Oxoid) at 37 °C, washed and diluted to 10^6^ or 10^4^ CFU/mL in ¼ Ringers solution (Oxoid) prior to inoculation (10 mL) in the M-SHIME stomach.

### 2.2. Human Fecal Donors

To assess the effect of human individual variability in resistance transfer, fresh fecal samples from six human volunteers (24–30 years old) were collected according to standard procedures. All volunteers followed a normal Western diet and had no history of gastrointestinal disorders nor antibiotic treatment 5 years prior to the sample collection. The fecal samples were homogenized and a 20% (w/v) fecal slurry in anaerobic phosphate buffer was prepared as described by De Boever [21]. All samples were processed and inoculated in the M-SHIME within 3 h after sample collection. Research incubation work with fecal microbiota from human origin was approved by the ethical committee of the Ghent University hospital under registration number BE670201836318.

### 2.3. Mucosal Simulator of the Human Intestinal Ecosystem (M-SHIME)

#### 2.3.1. General Setup 

The M-SHIME^®^-reactor setup (Prodigest and Ghent University, Belgium) was adapted to run eight M-SHIME runs in parallel (3 treatments + 1 control condition for 2 human donors). Each run consisted of a combined stomach-small intestine vessel and a proximal colon vessel (Figure 1). 

During reactor start-up, the colon vessels were inoculated with 25 mL fecal slurry and 475 mL nutritional medium (Prodigest NM002A containing 1.2 g/L arabinogalactan, 2 g/L pectin, 0.5 g/L xylan, 0.4 g/L glucose, 3 g/L yeast extract, 1 g/L special pepton, 2 g/L mucin, 0.5 g/L L-cystein-HCl and 4 g/L starch). A mucosal environment was created by adding 80 mucin type II agar-covered microcosms (AnoxKaldnes K1 carrier; AnoxKaldnes AB, Lund, Sweden) per colon vessel [22]. Every two days, 50% of the microcosms were replaced by new ones to mimic the natural renewal of the mucus layer. Three times a day and 90 minutes apart from each other, 140 mL acidified nutritional medium (pH2, HCl) and 60 mL digestive juice (12 g/L oxgall (BD)), 1.8 g/L porcine pancreatin (Sigma), 25 g/L NaHCO_3_ (Roth) were added to the stomach-ileum compartments. After 3 h the stomach-ileum content was pumped into the proximal colon vessels. Those vessel had a pH of 5.6–6.0, a volume of 500 mL and a retention time of 20 h. After a three-days start-up period, enabling the microbial community to adapt to the nutritional and physicochemical conditions in the colon vessels, the M-SHIME was inoculated with *E. coli* MB6212 (day 0, −3 h time point). To test whether *E. coli* donor strain concentrations effect the plasmid transfer ratios, two different *E. coli* MB6212 doses (10^5^ and 10^7^) were tested. To this end, 10 mL of bacterial suspension (either 10^6^ or 10^4^ CFU/mL) was inoculated in the stomach-small intestine vessels (final volume 210 mL) during feeding to obtain a final concentration of 4.8 × 10^4^ and 4.8 × 10^2^ CFU/mL respectively. The control stomach vessels were inoculated with 10 mL ¼ Ringers solution. After the inoculation, all stomach-ileum vessels were continued to be fed with 140 mL nutritional medium and 60 mL pancreatic juice three-times daily. During the whole experiment, anaerobic conditions were maintained by daily flushing with N_2_. The moment the inoculated *E. coli* strain entered the proximal colon vessels (i.e., 3 h after inoculation in the stomach vessels) was considered as the arbitrary 0 h time point (Figure 1).

At time points −72, −48, −24, 0, 2, 6, 24, 48 and 72 h, 10 mL lumen samples were taken and aliquoted for further analysis (cf 2.4. Analysis of lumen samples).

#### 2.3.2. Cefotaxime Treatment

The effect of cefotaxime treatment on resistance selection and transfer was determined by treating a stabilized M-SHIME with cefotaxime (Claforan, Sanofi-Aventis, Germany). Functional stability was assessed by SCFA measurements. Since cefotaxime is administered by intramuscular or intravenous injection, the M-SHIME was treated by cefotaxime containing mucin beads to mimic the natural diffusion through the colon wall. The recommended cefotaxime dose for an uncomplicated infection in adults is 2 g/day and approximately 10% ends up in the faeces [23]. Hence the estimated lumen concentration is 50 mg/L (200 mg/4L lumen). At time point 72 h of the M-SHIME run (Figure 1), a single dose cefotaxime was administered. Therefore, half of the mucin beads in colon vessel 4a and 4b were replaced with new ones containing 1250 µg cefotaxime/mL mucin suspension (20 mL mucine beads in 500 mL colon suspension = final concentration of 50 mg/mL cefotaxime), whereas those in the control vessels, i.e., 3a and 3b were replaced with regular mucin beads (total volume mucine beads: 20 mL. Assuming 100% diffusion of cefotaxime from the beads to the lumen, this leads to a final maximum concentration of 50 mg cefotaxime/L lumen. Four hours after replacing the beads, a meal (140 mL nutritional medium) was administered containing 10^7^ CFU *E. coli* MB6212. All colon vessels were covered with aluminum foil to shield the cefotaxime from daylight. Lumen samples were taken just before inoculation of *E. coli* MB6212 and 72 h after antibiotic administration. 

### 2.4. Analyses of Lumen Samples

#### 2.4.1. Bacterial Counting

Samples were serially diluted (¼ Ringers) and incubated on selective plates. MacConkey agar (nr3, Oxoid, 37 °C, aerobic) was used to count coliforms and Reinforced Clostridial Agar (RCA, Oxoid, 37 °C, anaerobic) for total anaerobic bacterial counts. Resistant [24] bacteria (transconjugants and the MB6212 donor strain) were counted on MacConkey and RCA with cefotaxime (0.25 µg/mL) in combination with sulfamethoxazole (256 µg/mL) as described by Lambrecht et al. [17]. Transconjugants on the antibiotic containing MacConkey agar were differentiated from MB6212 by their red morphology. To verify that their resistance was caused by p5876 and not due to intrinsic resistance or a spontaneous mutation, a PCR assay with primers targeting a specific and unique region on p5876 was performed [17]. On the antibiotic containing RCA plates, anaerobic transconjugants and the MB6212 *E. coli* could not be discriminated based on morphology. To estimate the number of true transconjugants among the anaerobic resistant colonies, 10 colonies per colon vessel were subcultivated on RAPID E’coli (to exclude MB6212) and analyzed by PCR.

#### 2.4.2. Short Chain Fatty Acids 

SCFA production was measured by capillary gas chromatography (GC-2014 gas chromatograph (Shimadzu, Hertogenbosch, the Netherlands) coupled to a flame ionization detector after diethyl ether extraction, as described by Andersen [25]. The total SCFA is the sum of acetate, propionate, butyrate, isobutyrate, valerate, isovalerate, caproate and isocaproate.

#### 2.4.3. Quantification of Total 16S rRNA Gene Abundance and of p5876 by qPCR

qPCR analysis was performed on a Lightcycler 480 Real-time PCR system (Roche). Total 16S rRNA gene abundance as a proxy for bacterial abundance and p5876 were quantified using SYBR Green technology. For each DNA extract (performed as reported by [26]), a 1000-fold dilution was made and analyzed in duplicate. QPCRs were carried out with eubacterial primes described by Ovreas [27] (Bac33FW: 5’ ACTCCTACGGGAGGCAGCAG and Bac518RV: 5’ATTACCGCGGCTGCTGG) and p5876-plasmid specific primers (FW 5’ GGCTGAGAAAGCCCAGTAAGG, RV: 5’TAAGTTGGCAGCATCACCTCG), designed in a former study [17]. The PCR program consisted of an activation step of 3 min at 98 °C, followed by 30 cycles of 10s at 98 °C and 30 s at 60 °C. Within each run, a standard curve was constructed using a 10-fold dilution series of 16S rRNA gene plasmid construct (IDT, Coralville, IA, USA) or a gblock fragment containing the p5876 targeted sequence to determine PCR efficiency [17]. 

#### 2.4.4. UHPLC-MS/MS 

Lumen samples taken just before and 72 h after cefotaxime treatment were analyzed using ultra high performance liquid chromatography-tandem mass spectrometry (UHPLC-MS/MS) to detect and quantify cefotaxime. 

For extraction, 4 mL lumen was spiked with ceftiofur-d_3_ (500 µg/L) and left to equilibrate at room temperature for 10 minutes. After addition of 6 mL acetonitrile, the sample was vortexed for 30 s. The sample was centrifuged (10 min, 1920 g) and the supernatant was collected and evaporated up to 4 mL under nitrogen at 40 °C. After filtration (0.22 µm) samples were loaded on the UHPLC.

The liquid chromatographic system consisted of an Acquity UPLC H-class system (Waters, Milford, MA). Separation was achieved on a Kinetex C18 column (2.1 mm × 100 mm, 1.7 µm, 100 Å) with a SecurityGuard ULTRA cartridge. The column was held at 35 °C, the injection volume was set at 5 µl and the eluent flow at 400 µL/min. The elution was performed with H_2_O:acetonitrile (95:5 v/v) + 0.05% acetic acid (solvent A) and gradually changing amounts of acetonitrile:H_2_O (95:5 v/v, solvent B).

The mass spectrometric equipment consisted of a Xevo TQ-MS (Waters) equipped with a Z-Spray system. Ions were generated using electrospray ionization in positive mode (ESI+). 

A calibration curve was generated by spiking a blanc lumen sample with different concentration of cefotaxime (0–1000 µg/L cefotaxime in MeOH:H_2_O) and ceftiofur-d3 (internal standard, 500 µg/L ceftiofur-d3 in methanol). The Limit Of Quantification (LOQ) is 10 µg/L.

### 2.5. Statistical Analysis of Bacterial Counts and SCFA Concentrations

All statistical analyses were performed in R (version 3.5.1). Linear mixed models were used for modelling the absolute and relative amount of cultivable resistant coliforms, anaerobes and SCFA. Sampling time points, total coliforms and anaerobes, MB6212 inoculation, and total SCFA concentrations were included as potential fixed effects. Human individual was considered as random factor to include inter-individual variability. 

All models were evaluated for normal distributed residuals with homogenous variance, by Shapiro Wilk test (*p* > 0.05) and visually by Q-Q plots. When a significant concentration effect was present (ANOVA, *p* < 0.05), the categories were compared pair-wise by posthoc analysis using Tukey’s test.

### 2.6. S Amplicon Sequencing and Bioinformatics Analysis

The lumen microbial community at time points 0 and 72 h and after cefotaxime administration was assessed using Illumina next generation 16S rRNA gene amplicon sequencing. DNA extraction was performed as reported by [28]. Library preparation and sequencing on a Illumina MiSeq platform occurred by LGC Genomics. The 341F (5’-CCTACGGGNGGCWGCAG-3’) - 785Rmod (GACTACHVGGGTATCTAAKCC) primer pair, derived from Klindworth et al. [29] was used to amplify the 16S rDNA V3-V4 region.

The Mothur software package (v1.39.5, [30]) was used to process Illumina data. Forward and reverse reads were assembled into contigs and ambiguous contigs or contigs with divergent lengths were removed. The number of unique sequences was determined and these were aligned to the Mothur-reconstructed SILVA Seed alignment (v123). Sequences not aligning within the region targeted by the primer set or sequences with homopolymer stretches with a length higher than 12 were removed. Sequences were pre-clustered together within a distance of 1 nucleotide per 100 nucleotides. These cleaned-up and preclustered sequences were checked for Chimera’s (with Uchime) [31] and classified using RDP v. 16 and a naive Bayesian classifier (Wang’s algorithm). Sequences classified as Eukaryota, Archaea, chloroplasts and mitochondria and those that could not be classified at all (even not at (super)kingdom level) were removed. Clustering of operational taxonomic units occurred with an average linkage and at 97% sequence identity.

Sequence data has been submitted to the NCBI database under accession number PRJNA579267.

Differences in microbial community composition between samples were explored using RC(M) ordination with reactor run as conditioning factor (fit Row-Column association Models, RCM package v0.1.0, [32]). Significant differences were identified by means of Permutational Multivariate Analysis of Variance (PERMANOVA) using the adonis function (vegan v.2.5-2, PairwiseAdonis [33] with human as strata, 1000 permutations, OTU level. Multivariate homogeneity of dispersion (variance) was calculated with the betadisper function (vegan). Absolute counts were used to determine Hill numbers (H0, H1 and H2, Phyloseq (v1.24.2)) to evaluate the microbial community structure. A Wilcoxon (Signed) Rank Sum test was used to detect significant differences in Hill numbers between antibiotic treated vessels and controls.

## 3. Results

### 3.1. Transfer of Resistance Plasmid p5876: Effect of Dose and Human Individual in the Absence of Antibiotics

To evaluate whether the dose of ingested antibiotic resistant *E. coli* and the individual gut microbiota affect the conjugal transfer of antibiotic resistance genes towards gastrointestinal microbiota, an M-SHIME system was set up. This system was fed with a nutritional medium containing either 10^5^ or 10^7^ antibiotic resistant *E. coli* MB6212, harboring the transferable resistance plasmid p5876. The emergence of resistant gut microbiota was monitored over time. Several M-SHIME setups were run to simulate the gastrointestinal system of 6 different human individuals. 

Transfer of the resistance plasmid p5876 was quickly observed, as after 2 h residing in the simulated colon, resistant, indigenous coliforms could be detected in the lumen of all human donors except those of human donor B (Figure 2). For the latter, resistant coliforms were detected only after 6 h or 24 h when inoculated with 10^7^ or 10^5^ CFU MB6212 respectively. Inoculation dose (10^5^ or 10^7^ CFU) had no significant effect on the absolute (Figure 2 and Figure 3) and relative number (Appendix A, Figure A2) of resistant indigenous coliforms. Yet, the presence of the *E. coli* MB6212 donor strain in the simulated proximal colon did result in a significant increase of resistant indigenous coliforms over time (Figure 3), with final resistance levels at 72 h ranging from 7.6 × 10^4^ (human A) up to 8.4 × 10^6^ (human C) CFU/mL (Figure 2). The *E. coli* MB6212 donor strain itself persisted in the proximal colon during the whole experiment as was demonstrated by the presence of pink colonies on RAPID’Ecoli agar plates. At the final sampling time point (72 h), 2.3 × 10^8^ ± 1.0 × 10^7^ CFU/ mL (mean ± SE) *E. coli* MB6212 were detected, independent of the administered dose or human individual (*p* > 0.05). 

Resistant anaerobes were detected in the control and inoculated proximal colon vessels. Based on colony morphology on selective RCA plates with antibiotics, no clear distinction could be made between resistant anaerobic transconjugants, *E. coli* donor strain and intrinsically resistant anaerobic bacteria. Hence, for anaerobes, statistics were performed on the total number of resistant anaerobes. The human individual, sampling time point, and MB6212 inoculation had no significant effect on the absolute and relative number of resistant anaerobes. On average 1.1 × 10^8^ CFU/mL lumen (± SE 2.4 × 10^7^ CFU/mL) cultivable resistant anaerobes were detected. 

Inoculation of 10^5^ (+ *E. coli* E5) or 10^7^ (+ *E. coli* E7) CFU of the resistance plasmid donor strain *E. coli* MB6212 in the stomach vessel at time point minus 3 h. Control: without inoculation of *E. coli* MB6212. Time: residence time in the proximal colon, ND: not detected, detection limit = 100 CFU/mL.

Resistant transconjugant coliforms and anaerobes were screened for the presence of p5876 by PCR. 94% (99/105) of the tested purple colonies on MacConkey and 62% (61/99) of the selected colonies on RCA tested positive for presence of the p5876 resistance plasmid. Based on its sequence, p5876 may confer resistance to sulfonamides (sul3), β-lactams bla-SHV2 and bla-pse4), chloramphenicol (cmLA), aminoglycosides (aadA1 and aadA2), tetracyclines (tetR, tetA), trimethoprim (dhrfI). In addition, several toxin-antitoxin genes involved in plasmid addiction are present: PemK, PemI, pndC, yacB, yacA (Appendix A).

The total amount of p5876 in the simulated proximal colon vessels was quantified by qPCR. At time point 0h in both control and inoculated vessels and at 72 h for control vessels no p5876 could be detected. The inoculated vessels at time point 72 h did however display an average 2 × 10^8^ p5876 copy number/mL. No significant differences in p5876 copy number were observed between different individuals and inoculation doses. In addition, no significant differences in total bacterial load—as measured by the 16S rRNA gene copy number—were detected between time points (0 h vs 72 h), individuals, and inoculated versus control vessels. Overall, on average 2.3 × 10^10^ 16S rRNA gene copy numbers/mL lumen (±8.3 × 10^9^, SE) were detected.

At each sampling time point, SCFA were measured as a marker for metabolic activity. Sampling time point and inoculation of the *E. coli* MB6212 donor strain had no significant effect on the total SCFA, nor on the separate butyrate, acetate and propionate levels. Inter-individual differences in SCFA profiles were observed with mean acetate:propionate:butyrate ratios of 42:33:24 (person A), 72:21:7 (person B), 67:24:9 (person C), 70:17:13 (person D), 66:21:13 (person E), 63:23:14 (person F).

The bacterial community composition and structure of the proximal colon lumen was explored with 16S rRNA gene amplicon sequencing. Inter-individual differences in dominant taxa were observed (Figure 4). In general, no significant differences in richness (OTU level, Hill number H0) nor in diversity (H1, H2) were detected between individuals. Within an individual, sampling time point (0h vs 72 h), inoculation of *E. coli* MB6212 (controls vs 10^5^ and 10^7^) and its dose (10^5^ vs 10^7^) had no significant effect on the Hill numbers.

To study differences between genus level composition of the lumen communities, taking into account the relative abundances, samples were ordinated in a two-dimensional plot using RC(M) analysis (Figure 5) with reactor run as confounding factor. Most samples of the same individual clustered closely together. 

### 3.2. Transfer of Plasmid p5876 after Cefotaxime Treatment

In the second part of this study, the transfer of p5876 under the selective pressure of a cefotaxime treatment was assessed. *E. coli* MB6212 (10^7^ CFU/mL) were inoculated in the stomach vessel immediately after cefotaxime treatment ([max. final lumen]: 50 mg/L). The cefotaxime concentration in the lumen was measured by UHPLC-MS/MS. However, 72 h after the treatment, no cefotaxime could be detected in the lumen anymore. 

Cefotaxime treatment led to a significant reduction (85%) in resistant coliform. The treatment had no significant effect on the total number of cultivable coliforms nor on the cultivable anaerobes (at time point 72 h) nor on the number of resistant anaerobes (data not shown). 

To assess the co-impact of cefotaxime treatment and MB6212 inoculation on SCFA production and the human gut microbiome composition, samples at time point 0h (just before inoculation and treatment) and at 72 h were analyzed (Figure 6). Inter-individual differences in SCFA profiles were observed, but overall, cefotaxime treatment caused a 38% decrease in total SCFA concentration (MB6212 vs MB6212 with cefotaxime, *p* = 0.03). In particular, a significant decrease in butyrate (60% reduction, *p* = 0.03) and propionate concentration (84% reduction, *p* = 0.002) was observed. No significant changes could be detected in acetate production.

Neither MB6212 nor cefotaxime treatment induced a clear long-term shift in community composition on family level (Figure 7). After 72 h, no significant differences in Hill numbers (H0, H1 and H2) between (i) controls, (ii) MB6212 inoculated and (iii) MB6212 inoculated and cefotaxime treated samples was observed. An RC(M) ordination of the luminal microbial community at genus level indicated that human individual and antibiotic treatment are the main factors determining grouping of the samples (Figure 5). *Veillonella* were more abundant in cefotaxime treated samples than in the others. On the other hand. *Desulfovibrio*, *Coprococcus* and *Rhodospirales* were more characteristic for non-cefotaxime treated samples. The distinction between cefotaxime treated and control communities was confirmed by a permanova (*p* < 0.05). 

## 4. Discussion

Inoculation of *E. coli* MB6212, harboring the p6785 resistance plasmid, in an M-SHIME revealed rapid transfer of this plasmid towards indigenous coliforms and anaerobes in the simulated proximal colon lumen. After 2 h residing in the colon vessels, transfer could already be observed, with resistance ratios ranging from 2.2 × 10^−5^ to 0.17 resistant indigenous coliforms/total coliforms. Resistance ratios could not be calculated for anaerobes as colony morphologies did not allow to differentiate between donor strain, transconjugants and intrinsic resistant anaerobes. Coliform resistance ratio’s require careful interpretation as they do not allow to distinguish between (i) direct transfer of the resistance plasmid from MB6212 to indigenous acceptor bacteria, (ii) second level plasmid transfer between indigenous transconjugants and acceptors and (iii) vertical transfer among coliforms during cell division. Hence, the resistance ratio only gives a rough indication of the conjugation efficiency. Moreover, in the current study transfer was studied by counting transconjugant colonies on selective plates with antibiotics and this is likely to be an underestimation of the real transfer ratio in the M-SHIME. Particularly because acquired resistance genes are not always functionally expressed in the new host, which could be either caused by conflicting codon usage, posttranslational modification and protein folding, or due toenvironmental conditions do not support the expression and/or growth of the bacterium. The plasmid acceptors in the M-SHIME setup were expected to be coliforms, which justifies the plating on MacConkey agar. The p5876 is an F-plasmid: these plasmids are among the most studied conjugative plasmids and are generally characterized by a high conjugation rate and narrow host range [34]. 

The worrisome and quick spread of p6785 in the simulated proximal colon indicates that this dense and nutrient rich environment favors horizontal plasmid transfer. In general, conjugation ratios have been shown to be highly variable as they depend on plasmid factors, but also on mating-pair formation relying on direct contact between donor and acceptor and on environmental factors. Environmental conditions, such as temperature [35], high cell density, glucose [36], quorum sensing [37] and (micro)aerobic conditions [38] have been shown to have an effect on the transfer of F-like plasmids.

The M-SHIMEs from human A, B and C harbored cefotaxime and sulfamethoxazole resistant coliforms in the control vessels, but tested negative for the presence of p6785 by colony PCR and qPCR on lumen samples. This indicates the presence of resistant bacteria in the initial fecal inoculum from these persons. Positive human donors are not surprising, since it has already been demonstrated in 1988 that in 62.5% of fecal samples from people without a recent antibiotic history, at least 10% of the culturable isolates on MacConkey were resistant to a single antibiotic and over a third was multiresistant [39]. In addition, several anaerobes including enterococci [40], bifidobacteria [41] and lactic acid bacteria [42] are known to be (intrinsically) resistant towards cefotaxime, tetracycline or sulfamethoxazole. This may explain the presence of resistant indigenous anaerobes in all tested humans (A–F) in the current study.

The results acquired by plating, PCR and qPCR indicate that transfer and persistence of the resistance plasmid can occur in the absence of antibiotics. Persistence of the p5876 plasmid in the M-SHIME implies that the plasmid is either stably segregated upon cell division, confers a fitness advantage to its host, harbors an addiction system, or is being transferred at very high rates. Since transfer and persistence also occurred in the absence of antibiotics, it is less likely that the resistance genes were the main drivers for plasmid conjugation and maintenance. As only 57% of the plasmid genes could be assigned to a function, it is possible that one of the undefined genes offers a fitness advantage to its host under M-SHIME conditions. Moreover, evidence of addiction systems – also called post-segregational killing systems were found on the plasmid sequence. 

As expected, all six M-SHIME setups had a unique 16S rRNA gene profile. Ingestion of the *E. coli* strain—either with 10^5^ or 10^7^ CFU/mL—did not result in a detectable shift in community composition at OTU level. In the current study, two *E. coli* donor concentrations were tested. Although it is believed that even very low *E. coli* concentrations may be able to colonize the gut and potentiate horizontal gene transfer [43,44], the lowest concentration tested in the M-SHIME was 10^5^ CFU/mL. Lower concentrations would be too difficult to detect by agar plating as the background microbiota would be too dense. In this study, an *E. coli* strain was chosen as a plasmid-donor, since *E. coli* are regarded as indicator organisms for tracking microbial resistance [45]. However, one should take into account that humans are exposed—through food or diverse environments—to many other commensal resistant bacteria, which harbor narrow/broad host range plasmids.

In the second part of this study, the effect of a single cefotaxime treatment on the p5876 plasmid transfer was evaluated. In general, antibiotic treatment is expected to reduce bacterial diversity, to expand or collapse specific taxa, to select for resistant bacteria as well as to increase the opportunity for horizontal transfer [46]. Cefotaxime diffusion into the lumen of the proximal colon (final concentration max. 50 mg/L lumen), simulating treatment of a non-complicated infection, resulted in an unexpected decrease in resistant coliforms. Potential plasmid acceptors are likely to be killed or inhibited before plasmid transfer could occur. The lowest Minimal Inhibitory Concentration (MIC) for cefotaxime is 2 µg/L, whereas the Predicted No Effect Concentration (PNEC) for cefotaxime resistance selection was set to 0.125 µg/L [47]. These values are assessed in pure cultures by lab cultivation and can differ from those under in vivo conditions which includes eg. bacterial cross-protection, degradation and biofilm formation. This might explain why the *E. coli* donor strain (MIC = 4 µg/mL) could still be recovered after treatment with a high cefotaxime concentration. Moreover, the cefotaxime decreased over time due to dilution which is inherent to the M-SHIME reactor dynamics and biotic and the abiotic breakdown [48]. 

In contrast, the single dose of cefotaxime had no long-term effect on the total amount of cultivable anaerobes nor on the resistant anaerobes. This is in concordance with the finding that the fecal inoculum of the human volunteers, and consequently also the control vessels, already harbored cefotaxime resistant anaerobes at the start of the M-SHIME experiment. 

Treatment with a single dose of cefotaxime caused a significant disturbance in bacterial metabolic activity, indicated by a reduction in total SCFA. This reduction was mainly due to a decrease in butyrate and propionate production, which might be unfavorable for the host if these low levels are remained. 

Cefotaxime treatment did not lead to a detectable shift in community richness and evenness (Hill numbers), probably because of the rapid diffusion and clearance of cefotaxime and because lumen samples were only taken 3 days after antibiotic treatment. RCM ordination analysis revealed that human individual and antibiotic treatment are the main grouping factors for the simulated lumen samples. *Veillonella* spp. were more abundant in cefotaxime treated samples. These Gram-negative anaerobic cocci, are known for their lactate fermenting capacities. As of yet, no Epidemiological Cut-off values for cefotaxime resistance for *Veillonella* are available, making it challenging to hypothesize about the rise of this genus in antibiotic treated samples. The increased Veillonella abundance was not visible in the 16S rRNA gene family level bar chart, suggesting that the increase is counteracted by other Veillonellaceae members. 

The current study was performed in a controlled M-SHIME system and while this system was built to mimic the in vivo gut conditions as good as possible and to allow careful monitoring and uncomplicated sampling, one of its major drawbacks is the lack of host cells. A previous study demonstrated that the immune system can enhance plasmid transfer under inflammation conditions by triggering blooms of certain resident bacteria, such as enterobacteria. [49]. On the other hand, human gut epithelial cells have been shown to reduce bacterial conjugation by secreted factors [50]. 

## 5. Conclusions

Transfer of resistance plasmids to our gut microbiota and colonization of resistant bacteria is an alarming scenario, leading to a gut antibiotic resistance reservoir involved in spreading resistance genes to transient or colonizing bacteria and pathogens entering our gut. This study demonstrated rapid transfer of a resistance plasmid from a commensal *E. coli* to indigenous coliforms. Transfer occurred in all six simulated human colons and was independent of the ingested dose (10^5^ vs. 10^7^ CFU).

## Figures and Tables

**Figure 1 life-11-00192-f001:**
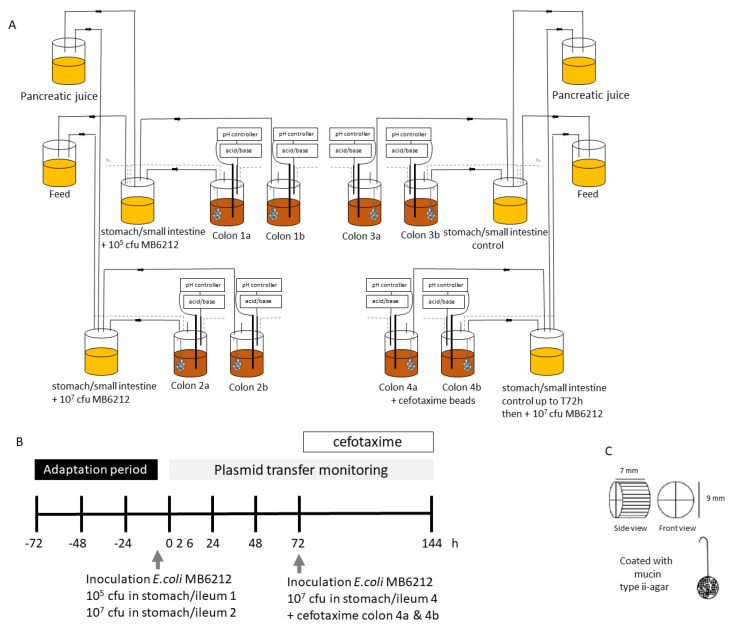
M-SHIME. (**A**): Reactor vessel setup. Two human individuals were tested in parallel (colon a,b) (**B**): Time frame of the experiment. −3 h: time point at which *E. coli* MB6212 is inoculated in the stomach vessel. 0 h: time point at which *E. coli* MB6212 enters the colon vessel, (**C**): Mucin microcosms.

**Figure 2 life-11-00192-f002:**
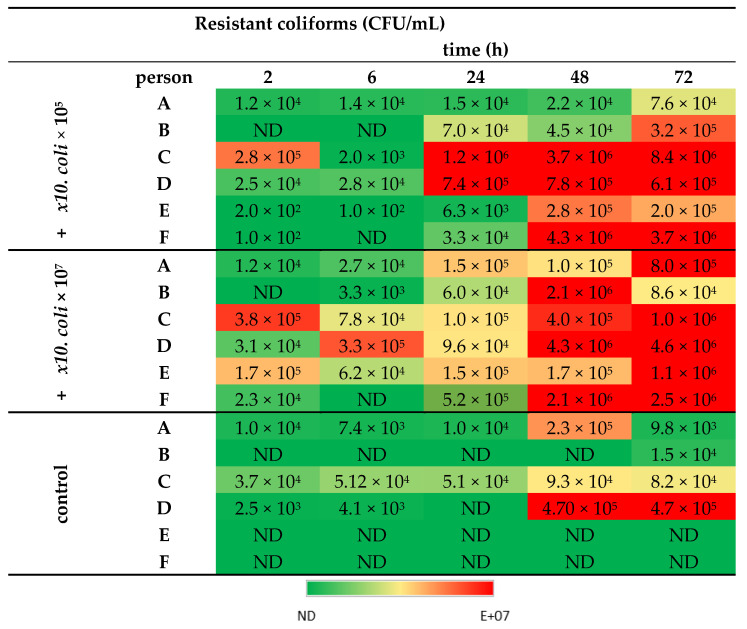
Emergence of resistant coliforms in the proximal colon lumen of 6 different human individuals (A–F) over time.Absolute numbers of resistant coliforms in the proximal colon (i.e., non-MB6212, cefotaxime + sulfamethoxazole resistant).

**Figure 3 life-11-00192-f003:**
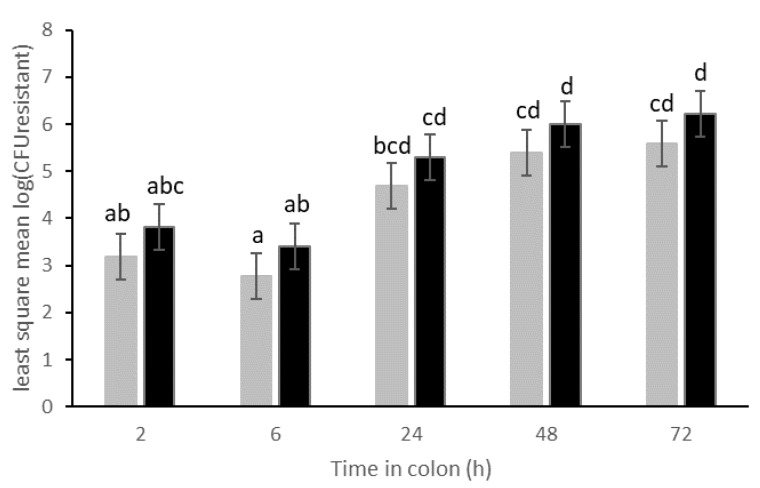
Least mean squares of the number of resistant coliforms (log CFU/mL) in the proximal colon vessels over time. Stomach vessels were inoculated with 10^5^ (grey bars) and 10^7^ (black bars) *E. coli* MB6212 donor strain at 0 h. Resistant transconjugants (non-MB6212) were enumerated by plating. Time points sharing a letter are not significantly different (*p* > 0.05) in post hoc analysis. Least mean squares ± standard error, n = 6 human donors.

**Figure 4 life-11-00192-f004:**
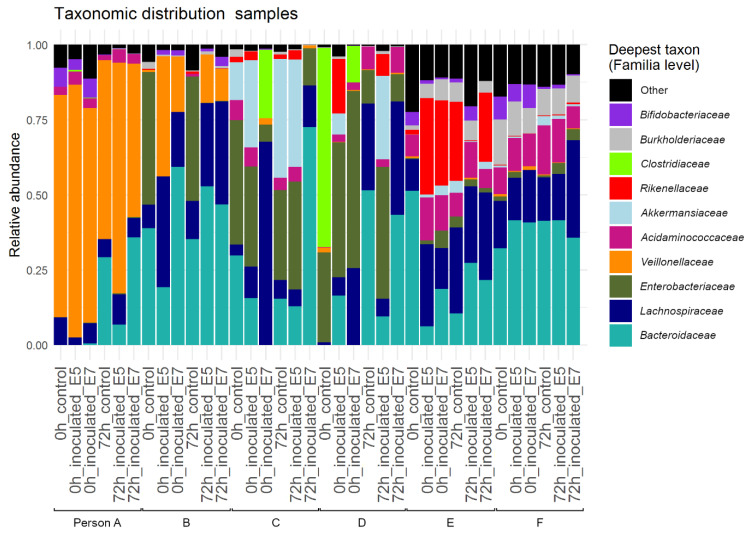
Relative abundance of the 10 most abundant families of the simulated luminal proximal colon microbial community based on Illumina sequencing of the V3-V4 16S rRNA gene fragment.

**Figure 5 life-11-00192-f005:**
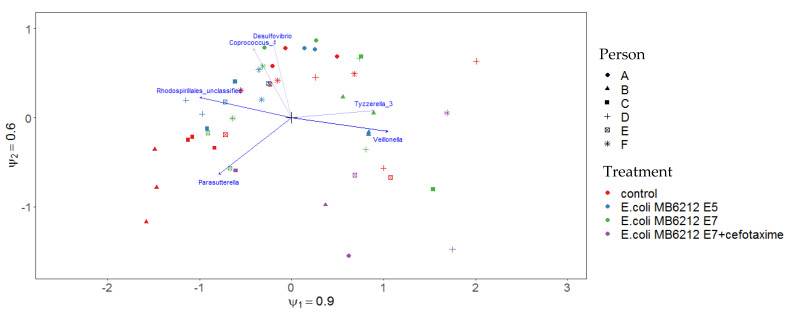
Biplot of the unconstrained RC(M) ordination describing the lumen microbial community composition at genus level, as determined by Illumina sequencing of the V3-V4 16S rRNA gene fragment.Only the six taxa that react most strongly to the environmental gradients are shown. Taxa are more abundant than average in samples to which their arrow points. Samples close together have similar taxon composition.

**Figure 6 life-11-00192-f006:**
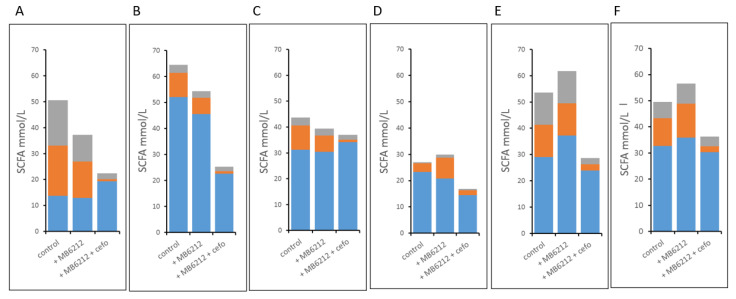
Main Short Chain Fatty Acid profiles of lumen samples from the proximal colon vessel of the M-SHIME of 6 human individuals (**A**–**F**). Grey: butyrate, orange: propionate, blue: acetate Samples were analyzed 72 h after inoculation of *E. coli* MB6212 + cefo: single dose of cefotaxime treatment through mucin-bead diffusion (50 mg/L). Control: without *E. coli* MB6212, without cefotaxime.

**Figure 7 life-11-00192-f007:**
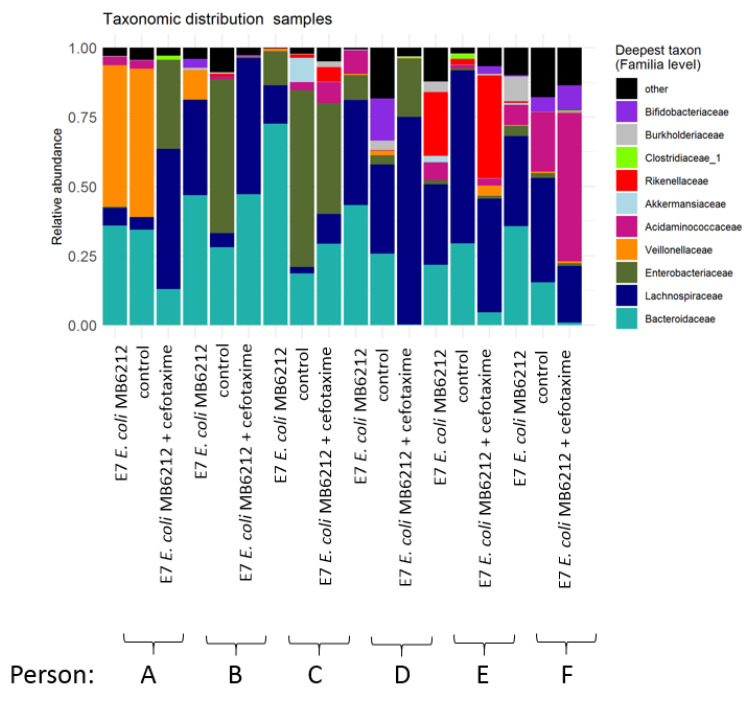
Relative abundance of the 10 most abundant families of the luminal microbial community based on Illumina sequencing of the V3–V4 16S rRNA gene fragment. Lumen samples from M-SHIME proximal colon vessels from 6 human individuals (person A-F) were analyzed 72 h after cefotaxime treatment [50 mg/L]. E7 *E. coli* MB6216: with inoculation of 10^7^ CFU cefotaxime resistant *E. coli* MB6212 in the stomach vessel, control: without cefotaxime treatment and without *E. coli* MB6212 inoculation.

## Data Availability

Sequence data has been submitted to the NCBI database under accession number PRJNA579267.

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
