# Peer review of "Transfer of Antibiotic Resistance Plasmid from Commensal E. coli towards Human Intestinal Microbiota in the M-SHIME: Effect of E. coli dosis, Human Individual and Antibiotic Use"

_life, 2021, doi:10.3390/life11030192_

Round 1
Reviewer 1 Report
This is a well written paper addressing the effect of dose on plasmid transfer, in the absence and in the present of an antibiotic, using a nice system mimic the human gut. The authors find rapid transfer of plasmid p5876 and no significant effect of the doses that were tested in the absence of antibiotic. The authors also show that cefotaxime reduced the transfer and the concentration of short chain fatty acids. The paper makes a nice discussion of the results and the pros and cons of the study system.
Some minor issues should be addressed:
Line 46, please detail what kind of in silicons analysis: comparative genomic analysis?
In the introduction, some more citations are need: Line 65, please provide citation for "hotspot for gene transfer"; Line 68, please provide citation that shows Mobile ARGs are significantly enriched in proteobacteria;
Line 102: how was the strain made lac-? What is the genetic basis of the phenotype?
Line 115: It is not clear if the study was ethically approved.
Line 158: clarify what a stabilised M-SHIME is and how was stability assessed.
Figure 2: quality should be improved, in particular the size of the dots should be increased and error bars added.
Line 421: typo 2.3x10^10.
Line 499: some typo 10^7
Line 602: Cite PMID: 31431529 for a nice example of HGT in E. coli.
Author Response
Please see the attachement

Reviewer 2 Report
The use of a controlled M-SHIME system to assess the transfer of resistance plasmids to human intestinal microbiota is a very interesting and sophisticated approach. The experiments were carefully performed, and I must admit that I have enjoyed reading this nicely written manuscript. Please see below only for a few minor comments:
1. I personally think that the use of a heatmap would be better suited for displaying Figure 2, for visualisation and comparison of the presence and the abundance level of resistant coliforms among different individuals at various time points, plus three different setups. For the current figure, I find that it is rather easy to lose track of one individual’s samples when toggling between different time points and treatment conditions.
2. Line 62-64: replace “-” with “,”
3. Line 162: it is better to use the word “approximately” than c. which is usually used for dates.
4. Lines 563-564: revise “- a cause…conditions” to “, which could be either caused by conflicting codon usage, posttranslational modification and protein folding, or due to unfavourable environmental conditions”
